DOI: 10.1038/s41467-018-04638-2　　**OPEN**

# An ensemble code in medial prefrontal cortex links prior events to outcomes during learning

Silvia Maggi [1,2], Adrien Peyrache [3] & Mark D. Humphries [1,2]

The prefrontal cortex is implicated in learning the rules of an environment through trial and error. But it is unclear how such learning is related to the prefrontal cortex's role in short-term memory. Here we ask if the encoding of short-term memory in prefrontal cortex is used by rats learning decision rules in a Y-maze task. We find that a similar pattern of neural ensemble activity is selectively recalled after reinforcement for a correct decision. This reinforcement-selective recall only reliably occurs immediately before the abrupt behavioural transitions indicating successful learning of the current rule, and fades quickly thereafter. We could simultaneously decode multiple, retrospective task events from the ensemble activity, suggesting the recalled ensemble activity has multiplexed encoding of prior events. Our results suggest that successful trial-and-error learning is dependent on reinforcement tagging the relevant features of the environment to maintain in prefrontal cortex short-term memory.

[1] School of Psychology, University of Nottingham, Nottingham NG7 2RD, UK. [2] Faculty of Biology, Medicine, and Health, University of Manchester, Manchester M13 9PT, UK. [3] Montreal Neurological Institute, McGill University, Montreal H3A 2B4, Canada. Correspondence and requests for materials should be addressed to M.D.H. (email: mark.humphries@nottingham.ac.uk)

Learning the statistical regularities of an environment requires trial and error. But how do we know what is relevant in the environment in order to learn its statistics? In other words: how do we know what to remember? It seems likely that medial prefrontal cortex plays a role here[1, 2]: it is needed for trial and error learning of correct behavioural strategies[3–5], neuron and neural ensemble activity represents abstract and context-dependent information related to the current strategies[6–9], and changes to ensemble activity correlate with shifts in learnt behavioural strategies[10–12]. Moreover, medial prefrontal cortex receives a direct projection from the CA1 field of the hippocampus that may allow the integration of spatial information about the environment[13–17]. But medial prefrontal cortex also plays a role in short-term and working memory for objects, sequences, and other task features[17–25], upon which successful learning of statistical regularities may depend. It is unknown how relevant information about the statistics of the environment is tagged for memory in the medial prefrontal cortex.

An hypothesis we consider here is that reinforcement tags preceding choices and features to remember, in order to learn the rules of the environment. This hypothesis predicts that the reliable appearance of reinforcement-driven short-term memory activity in medial prefrontal cortex should precede successful learning. While previous studies have shown that prefrontal cortex activity patterns shift between rule changes[10, 11], including immediately before a shift in behaviour[12], none have looked at learning-driven changes to prefrontal cortex activity in the naive animal, nor what the activity encodes about the task.

To test this hypothesis, we analysed neural and behavioural data from rats learning new rules on a Y-maze. We took advantage of a task design in which there was a self-paced return to the start position of the maze immediately after the delivery or absence of reinforcement, yet no explicit working memory component to any of the rules. Consequently we could examine ensemble activity in medial prefrontal cortex during this self-paced return and ask whether or not a short-term memory encoding of reinforcement-tagged task features existed, in the absence of overt working memory demands.

Here we report that medial prefrontal cortex contains an ensemble code that links prior events to reinforcement. We show that a similar pattern of ensemble activity was specifically recalled after reinforcement and not after errors. This recall only reliably occurred in sessions with abrupt shifts in behavioural performance that indicated successful learning of a rule, and not during external shifts in reinforcement contingency, or in other task sessions. The recalled pattern appeared shortly before the abrupt shift in behaviour, and faded shortly thereafter, consistent with a causal role in learning. From the activity of the recalled ensemble, we could simultaneously decode retrospective task parameters and choices in a position-dependent manner, even from the naive animal. Together, these results show that learning was preceded by reinforcement-triggered ensemble activity that retrospectively and multiply encoded task parameters. They provide a link between the roles of medial prefrontal cortex in working memory and in rule learning, and suggest that reinforcement tags prefrontal cortex-based representations of choices and environment features that are relevant to trial and error learning of statistical regularities in the world.

## Results

**Step-like rule learning on a Y-maze**. In order to address whether and how medial prefrontal cortex neural activity encodes short-term memory during rule learning, we used medial prefrontal cortex population recording data previously obtained from a maze-based rule-learning task[26]. Four rats learnt rules for the direction of the rewarded arm in a Y-shaped maze, comprising a departure arm and two goal arms with light cues placed next to the reward ports (Fig. 1a). Each session was a single day with ~30 min of training, and 30 min of pre-training and post-training sleep. During training, the rat initiated each trial from the start of the departure arm; the trial ended when the rat arrived at the end of its chosen goal arm. During the following inter-trial interval the rat consumed any delivered reward, if the correct arm was chosen according to the current rule, and then made a self-paced return to the start position after consuming the reward, taking on average 70 s (67.8 ± 5.4 s, mean ± SEM) to complete the interval. Tetrode recordings from medial prefrontal cortex were obtained

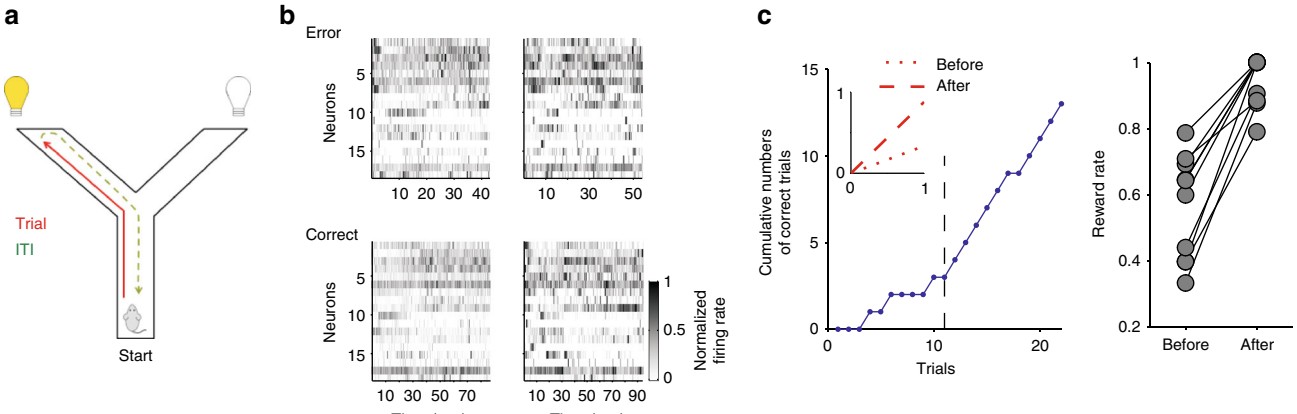

**Fig. 1** Task and learning sessions. **a** Schematic representation of the Y-maze. The trial starts with the animal at the start of the departure arm, and ends when it reaches the end of the chosen arm. The inter-trial interval (ITI) is a self-paced return back to the start position. **b** Examples of medial prefrontal cortex population activity during inter-trial intervals from the same session (two following errors and two following correct choices of arm). The heatmaps show the spike trains for all recorded neurons, convolved with a Gaussian of width $\sigma = 100$ ms. **c** Learning sessions contain abrupt transitions in performance. Left panel: Learning curve for one example learning session. The cumulative number of correct trials shows a steep increase after the learning trial (black dashed line), indicating the rat had learnt the correct rule. Inset: fitted linear regressions for the cumulative reward before (dotted) and after (dashed) the learning trial, quantifying the large increase in the rate of reward accumulation after the learning trial. Right panel: the rate of reward accumulation before and after the learning trial for every learning session (one pair of symbols per learning session; one session's pair of symbols are obscured). The rate is given by the slopes of the fitted regression lines

from the very first session in which each rat was exposed to the maze (Fig. 1b). Thus, the combination of a self-paced post-decision period—without experimenter interference—and neural activity recordings from a naive state allowed us to test for medial prefrontal cortex ensemble activity correlating with short-term memory during rule learning.

After achieving stable performance of the current rule, indicated by 10 contiguous correct choices, the rule was changed, unsignalled, in sequence: go right; go to the cued arm; go left; go to the uncued arm. Notably, none explicitly required a working memory component (such as an alternation rule). In the original study[26], the session in which initial learning of each rule occurred was identified posthoc as the first with three consecutive correct trials followed by 80% performance until the end of the session; the first of the initial three correct trials was identified as the learning trial. Ten sessions met these criteria, and are dubbed here the "learning" sessions. We first confirmed that these ten learning sessions showed an abrupt transition in behavioural performance (Fig. 1c), indicating the step-like change in behaviour commonly seen in successful learning of contingencies[10, 12, 27, 28]. In total, we examined 50 sessions, comprising 10 learning sessions, 8 rule change sessions, and 32 other training sessions (labelled "others" throughout).

**Reinforcement-driven recall of ensemble activity during learning**. We sought to identify signatures of short-term memory encoding by examining ensemble activity in the inter-trial intervals within each session. One signature of similar memory encoding between inter-trial intervals would be a repeated pattern of co-active neurons across the intervals, indicating a similar encoding. To allow comparisons between intervals, we thus first identified the core population of neurons in each session by selecting those that were active in every inter-trial interval. The proportion of recorded neurons retained in the core population was on average $74 \pm 2\%$ (SEM) across sessions (Supplementary Fig. 1). No clear difference in the size of this core population were observed between learning and any other session type (Supplementary Fig. 1). There was also no systematic recruitment or suppression of neurons not in the core population by learning or rule changes (Supplementary Fig. 1). Together, these results suggest that any potential short-term memory encoding specific to learning was not then simply a change in the proportion of active neurons.

We then asked if this core population contained a repeated pattern of co-active neurons between inter-trial intervals. We characterised the co-activity for each inter-trial interval by computing the pairwise similarity between the Gaussian-convolved spike trains of neurons in the core population (we use a Gaussian width of $\sigma = 100$ ms here, as in the example of Fig. 1b; the effects of varying $\sigma$ are detailed below). The pattern of co-activity in interval $t$ was thus described by a matrix $\mathbf{S_t}$ of pairwise similarities between neuron activity (Fig. 2a). To then quantify if a similar pattern of co-activity occurred between inter-trial intervals $t$ and $u$, we computed the similarity between their co-activation matrices $\mathbf{S_t}$ and $\mathbf{S_u}$ (Fig. 2b). Repeating this for all pairs of inter-trial intervals in a session gave us a matrix $\mathbf{R}$ of pairwise similarities between intervals, showing which inter-trial intervals had similar patterns of ensemble activity (Fig. 2c)—we dubbed this the recall matrix.

We found that patterns of neuron co-activity were more similar between intervals after correct trials than after error trials (Fig. 2c, d). We observed this preferential recall of ensemble activity following reinforcement in the majority of sessions (47/50 sessions; 37/50 had $P < 0.05$ for a Kolmogorov–Smirnov test between the distributions of recall values after correct and after

error trials—example distributions in Fig. 2c). This result would suggest that reward triggered a specific pattern of ensemble co-activity during the inter-trial interval. However, we were mindful that the inter-trial intervals following a correct trial were generally much longer than those following error trials (correct inter-trial intervals: $79.1 \pm 6.4$ s; error inter-trial intervals: $48.4 \pm 3.7$ s), because the animal lingered at the reward location (Supplementary Fig. 2). This difference in duration could systematically bias estimates of co-activity, simply because many more spikes would be emitted during intervals after correct choices than after errors. Thus, greater similarity between ensemble activity patterns for correct intervals could simply be due to more reliable estimates of the similarity between each pair of neurons.

To control for this, we used shuffled spike trains to compute the expected pairwise similarities between neurons due to just the duration of each interval; from these shuffled data similarity matrices per interval we then computed the expected recall matrix (Supplementary Fig. 3). Consequently, by subtracting this expected recall matrix from the data-derived recall matrix, we obtained a "residual" recall matrix: the similarity between ensemble activity patterns that remained after any effect of the durations of the inter-trials had been factored out (Fig. 2c). We used this residual recall matrix for all further analyses. With this correction, we still found that patterns of ensemble activity were more similar after correct trials than after error trials in the majority of sessions (34/50 sessions; 26/50 had $P < 0.05$ for a Kolmogorov–Smirnov test between the distributions of residual recall values after correct and after error trials). This result suggests that reward triggered a specific pattern of ensemble co-activity during the inter-trial interval.

We then examined how this reinforcement-driven recall of an ensemble activity pattern corresponded to the rats' behaviour (Fig. 2e). We found that only learning sessions had a systematically stronger recall of the same ensemble activity pattern after reinforcement (mean difference in recall: 0.072). Sessions in which the rule changed did not show a systematic recall after reinforcement (mean difference in recall: 0.042), ruling out external changes to contingency as the driver of the recall effect. Similarly, there was no systematic reinforcement-driven recall in the other sessions (mean difference in recall: 0.03), ruling out a general reinforcement-driven effect. When we further grouped these other sessions into those with evidence of incremental learning and those without, we still did not observe a systematic reinforcement-driven recall effect in either group (Supplementary Fig. 3). Finally, we found that the likelihood of obtaining ten systematically positive recall sessions by chance was $P < 0.01$ (permutation test). Together, these data show that a similar pattern of ensemble activity was only reliably recalled following reinforcement during the self-driven step-change in behaviour indicative of learning a rule.

The reinforcement-driven recall of an ensemble activity pattern could potentially be triggered only during reward consumption, and so not sustain across the return trip to the start position of the maze. To check this, we divided the neural activity into the activity occurring at the reward location and the activity during the return trip, and then separately computed the recall analysis on both. We found no systematic recall of a reinforcement-driven activity pattern at the reward location, and weak recall during the rest of the return trip (Supplementary Fig. 4), suggesting the reinforcement-driven recall of an ensemble activity pattern occurred throughout the inter-trial interval.

We asked how the recall of a pattern of ensemble activity was dependent on the temporal precision at which the similarity between neurons was computed. Here, this precision was determined by the width of the Gaussian convolved with the spike trains. We found that the reinforcement-driven recall of an

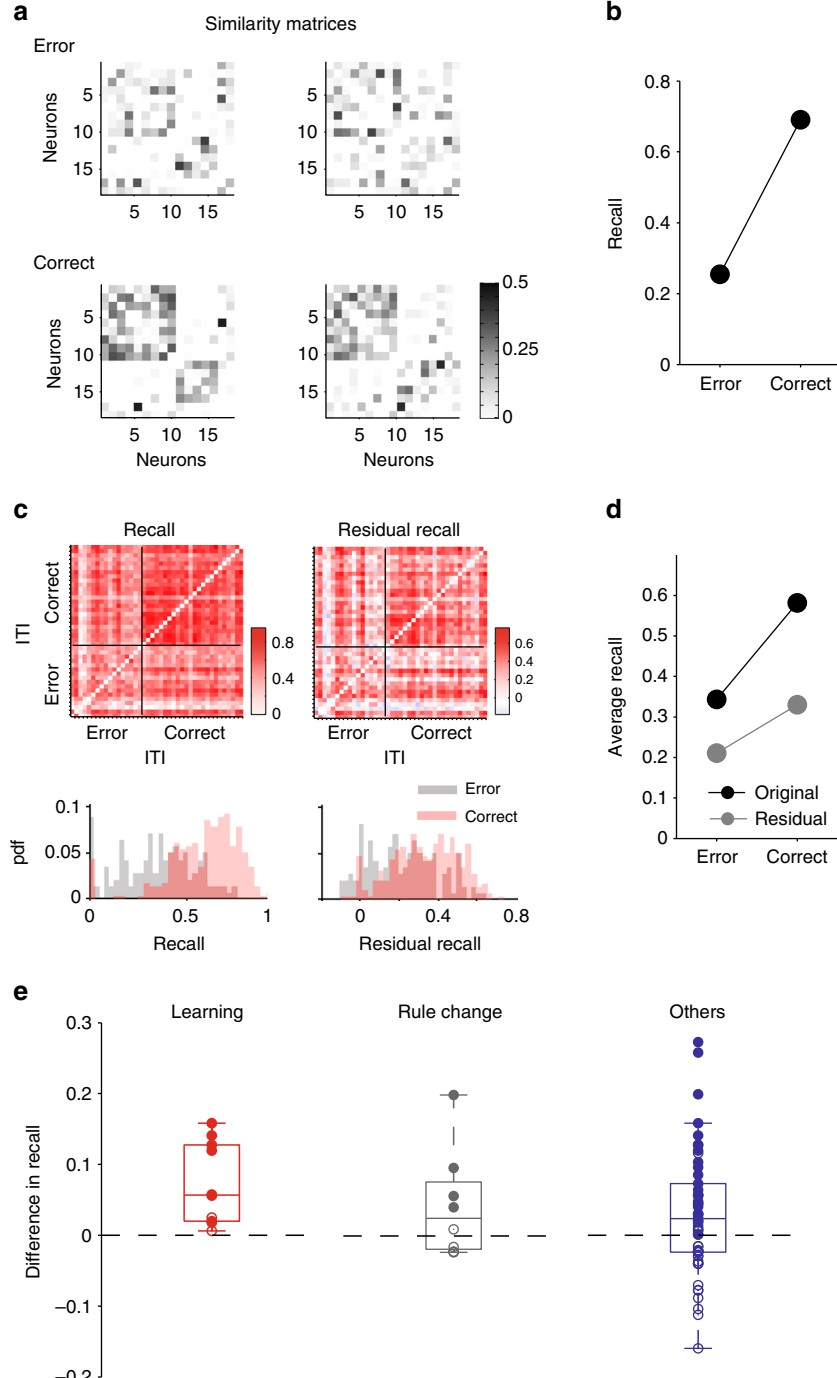

**Fig. 2** Outcome-selective recall of an ensemble activity pattern is learning-related. **a** Example similarity matrices for neural population activity during inter-trial intervals (same examples as Fig. 1b). **b** Example comparison of ensemble activity between inter-trial intervals. "Recall" $R(t,u)$ is the similarity between the core population's similarity matrices in intervals $t$ and $u$. For the matrices in panel a, the recall is lower following errors than following correct choices. **c** Example of consistent recall after reinforcement in one session. Left: the recall matrix **R** for the session, each entry the recall value $R(t,u)$ for inter-trials intervals $t$ and $u$. The recall matrix is ordered by the outcome of the preceding trial. Below we plot the probability density functions for the distribution of recall values, for the post-error pairs of intervals (bottom-left block diagonal in the recall matrix) and for the post-correct pairs of intervals (top right block diagonal in the recall matrix). Right: the residual recall matrix for the same session, after correction for the effects of interval duration. **d** The average recall values for post-error and post-correct intervals of the two matrices in **c**. The distribution of recall in the post-correct intervals was higher than in the post-error intervals (K–S test; recall: $P < 0.005$; residual recall: $P < 0.005$; $N(\text{correct}) = 24 \times 24 = 576$; $N(\text{error}) = 17 \times 17 = 238$.) **e** The difference in average residual recall between the post-correct and post-error intervals, sorted by session type. Each dot is one session. Filled circles indicate a positive difference at $P < 0.05$ between the distributions of recall values in the post-error and post-correct intervals (Kolmogorov–Smirnov test). Within each session type, two-sided sign tests that the median difference is not zero gave: learning, $P = 0.002$ ($N = 10$); rule change, $P = 0.72$ ($N = 8$); other sessions, $P = 0.37$ ($N = 32$)

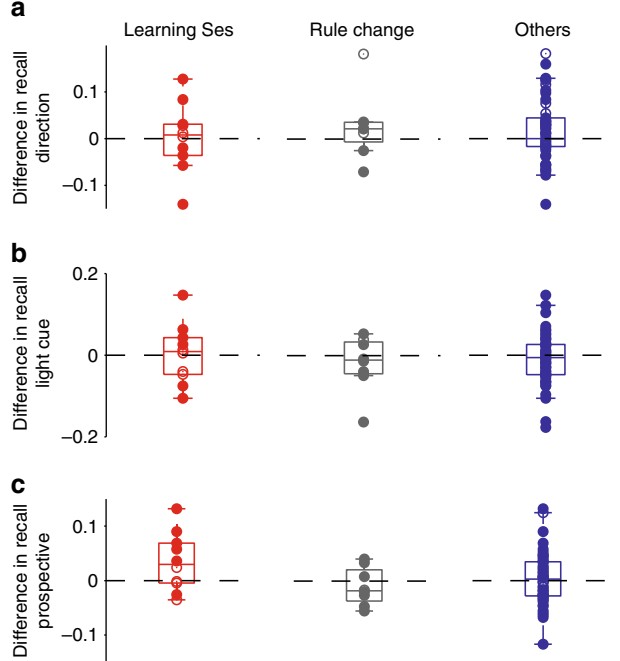

**Fig. 3** Recalled ensemble activity patterns are outcome-specific and encode retrospective outcome not future choice. **a** The difference in average recall between intervals after choosing the left arm or after choosing the right arm, sorted by session type. Filled circles here and in other panels indicate a significant difference between the distributions of recall values in the two sets of intervals (Kolmogorov–Smirnov test, $P < 0.05$). **b** As for **a**, but comparing intervals after the light cue appeared at the end of the left or the right arm. **c** The difference in average recall between intervals before error or before correct trials, testing for the prospective encoding of upcoming choice

ensemble in learning sessions was consistent across a wide range of Gaussian widths from 20 ms up to around 140 ms (Supplementary Fig. 5). Moreover, across the same range of Gaussian widths, we also consistently found that the recall effect for the learning sessions was greater than for rule change or other sessions (Supplementary Fig. 5).

**Recall of activity patterns is specific to prior reinforcement**. These results pointed to the hypothesis that, during successful rule learning, the reliable recall of a pattern of ensemble activity is triggered by prior reinforcement. To test this hypothesis, we first asked whether the recalled ensemble activity pattern was specifically triggered by reinforcement, and then whether it was specific to retrospective rather than prospective reinforcement.

To test if the recall was specifically triggered by reinforcement, we reorganised the residual recall matrix of each session by either the direction of the chosen goal arm (left/right) or by the cue position (left/right) on the previous trial. We found there was no systematic recall of ensemble activity patterns evoked by one direction over the other for either the chosen arm or the cue position (Fig. 3a, b). The systematic recall effect during learning thus appeared to be specific to reinforcement.

Variation in prefrontal cortex activity has however been linked to variations in behaviour independent of any reinforcement[23, 29]. Here, we sought behavioural differences between the intervals following error and correct trials that were not directly linked to reinforcement. We had already ruled out some such forms of behavioural variability: the use of residual recall throughout eliminates any effect of differences in elapsed time; and we have just seen there was no apparent preferential recall dependent on

the direction of origin (left or right arm). As a further test for the effects of variations in physical behaviour, we checked if variations in the length of the path back to the start could account for the recall effect, following previous demonstrations of trajectory effects on individual neurons in medial prefrontal cortex[23, 29]. We found that the distributions of path lengths did not consistently differ between error and correct trials, and that the learning sessions had no difference in recall between short and long paths, ruling out path length as an explanatory variable for reinforcement-driven recall (Supplementary Fig. 6). Intriguingly, when considering all sessions together, we saw that shorter path lengths correlated with stronger recall of ensemble activity (Supplementary Fig. 6f), possibly suggesting that the maintenance of short-term memory was more stable on more direct paths (independent of time elapsed). This finding suggests interesting future avenues for exploring behavioural correlates in prefrontal cortex; nonetheless, as path lengths in the learning sessions did not differ between error and correct trials (Supplementary Fig. 6g), this path-length correlate is orthogonal to the reinforcement-driven recall in learning sessions.

Modulation of medial prefrontal cortex activity by expected outcome or anticipation of reinforcement has been repeatedly observed[30–33], suggesting the recalled ensemble pattern could instead be a representation of the expected outcome on the next trial. To test if the recall effect was specific to retrospective reinforcement, we reordered the residual recall matrices according to the reinforcement received in the trial after the inter-trial interval. We found no systematic recall of an ensemble activity pattern in intervals preceding correct trials in any session type (Fig. 3c). In particular, for the learning sessions the systematic recall we observed for retrospective outcomes was not observed for prospective outcomes (compare Fig. 2c), and the magnitude of recall was larger for retrospective than prospective outcomes across all tested temporal precisions of similarity between spike trains (Supplementary Fig. 5d).

We were surprised that we could observe such a consistent difference between the retrospective and prospective recall in the learning sessions. By their nature, the learning sessions tend to be split into a sequence of error trials followed by a sequence of correct trials (cf Fig. 1c), so each trial outcome is frequently preceded and followed by the same type of outcome. Consequently, we create similar groups of "correct" and "error" intervals whether we choose to split intervals into groups by their following correct trials or by their preceding correct trials. Nonetheless, the systematically stronger retrospective recall across a wide range of timescales, despite the few error trials interspersed with correct trials, suggests that the recall of ensemble activity is dependent on prior, not future, reinforcement. (And as we show below, this conclusion is consistent with the complete absence of prospective coding of task elements by the ensemble's activity). Together, these results support the hypothesis that a specific pattern of ensemble activity triggered by just-received reinforcement appeared during successful rule learning.

**Recalled ensemble activity anticipates the behavioural transition**. This leaves opens the question of whether the appearance of this recalled ensemble pattern is a pre-condition of successful learning, or a read-out of already learnt information. If a pre-condition, then the recalled ensemble pattern should have appeared before the transition in behaviour indicating rule acquisition.

We thus sought to identify when the recalled ensemble activity pattern first appeared in each learning session. To look for this onset of the co-activity pattern, we put the recall matrix of each learning session in trial order (Fig. 4a). For each inter-trial interval in turn, we compared the strength of recall in the inter-

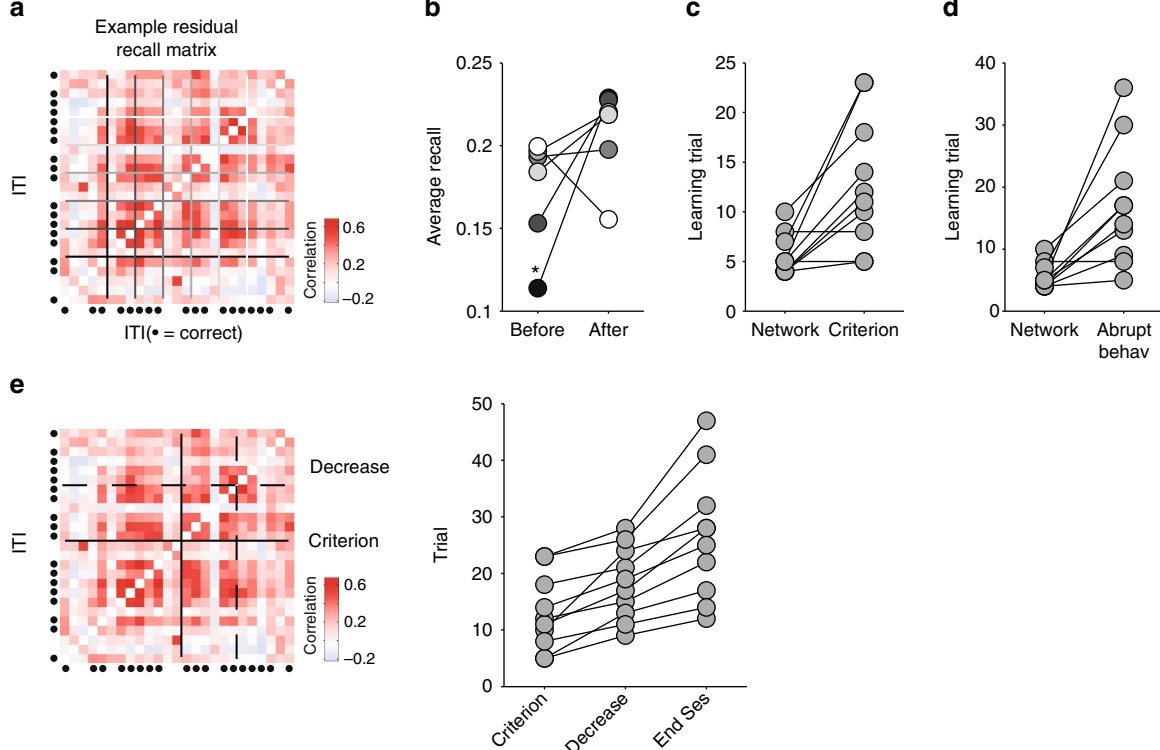

**Fig. 4** The recalled ensemble activity pattern anticipates behavioural learning. **a** A residual recall matrix in its temporal order for one example learning session. Columns are ordered from left to right as the first to last inter-trial interval (rows ordered bottom to top). For each inter-trial interval, the distributions of the recall values before and after the selected inter-trial interval were compared (Kolmogorov–Smirnov statistic: see Methods). Each grey scale line corresponds to the selected subset of the dividing inter-trial intervals plotted in **b**. **b** For each grey scale line in **a**, the corresponding average recall value before and after the dividing inter-trial interval. The asterisk indicates the inter-trial interval with the largest increase in recall after it, signalling the abrupt appearance of the recalled ensemble pattern. **c** Comparison of the learning trial identified by the original behavioural criterion with the identified onset trial for the recalled ensemble activity pattern ('Network'). **d** As **c**, but with the behavioural learning trial identified as the trial with the steepest change in the cumulative reward (Methods). **e** Testing for decay of the ensemble activity pattern. Left panel: example residual recall matrix in trial order for one learning session. The black solid line is the learning trial, while the dashed line is the identified offset of the recalled ensemble activity pattern. Right panel: For each learning session the learning trial (original criterion) is compared to the identified offset of the ensemble recall, and to the last trial of the session

trial intervals before and after that interval (Fig. 4b). We used the inter-trial interval corresponding to the largest difference in recall to identify the onset trial—the trial after which the ensemble activity pattern first appeared—as this indicated a step-increase in the similarity of activity patterns between inter-trial intervals.

We found that the recalled ensemble pattern appeared before or approximately simultaneously with the behavioural transition in all sessions (Fig. 4c, d). This was true whether we used the original behavioural criterion from ref. [26], or our more stringent definition of "abrupt" change in the cumulative reward curve (the trial corresponding to the greatest change in slope of the reward accumulation curve; see Methods). The timing of the appearance of the recalled ensemble pattern was thus consistent with it being necessary for successful rule learning.

As the change to the ensemble activity was often abrupt and so close to the behavioural change, this raised the question of what change to the underlying neural circuit drove this change in activity. One possibility would be a physical alteration of connectivity, forming a true "structural" cell assembly[34]. Alternatively, it could be a temporary effect, as might arise from a sustained change in neuromodulation[35, 36], forming a transient "functional" cell assembly.

To decide between these alternatives, we tested for the presence of a long-lasting physical change by assessing the longevity of the recalled ensemble activity pattern. Specifically, we tested whether

the recall of the ensemble was sustained until the end of the learning session by performing the onset analysis in reverse (Fig. 4e): for each inter-trial interval in turn, we checked whether the recall after that interval was significantly smaller than before it. We indeed found a statistically robust fall in the recall of the ensemble activity pattern in every learning session. A strict ordering was always present: the decay of the recalled ensemble was after the identified onset of recall, but before the end of the session (Fig. 4e), even though we did not constrain our analysis to this ordering. For the original set of identified learning trials, the decay trial was always after the learning trial (Fig. 4e). (If we used our alternative learning trial definition—the trial with the greatest change in reward accumulation—then 7 of the 10 sessions had decay after the learning trial, with 3 sessions showing decay before it). This analysis indicates the recalled ensemble activity pattern formed transiently during learning, and decayed quickly after learning was established.

**Inter-trial interval ensembles were not recalls of trial ensembles**. We then turned to understanding what the recalled activity pattern encoded. One possibility is that the recalled ensemble was just the pattern of ensemble activity in the preceding trial, reflecting some replay of the pattern of correlated activity that preceded (and possibly caused) a correct choice.

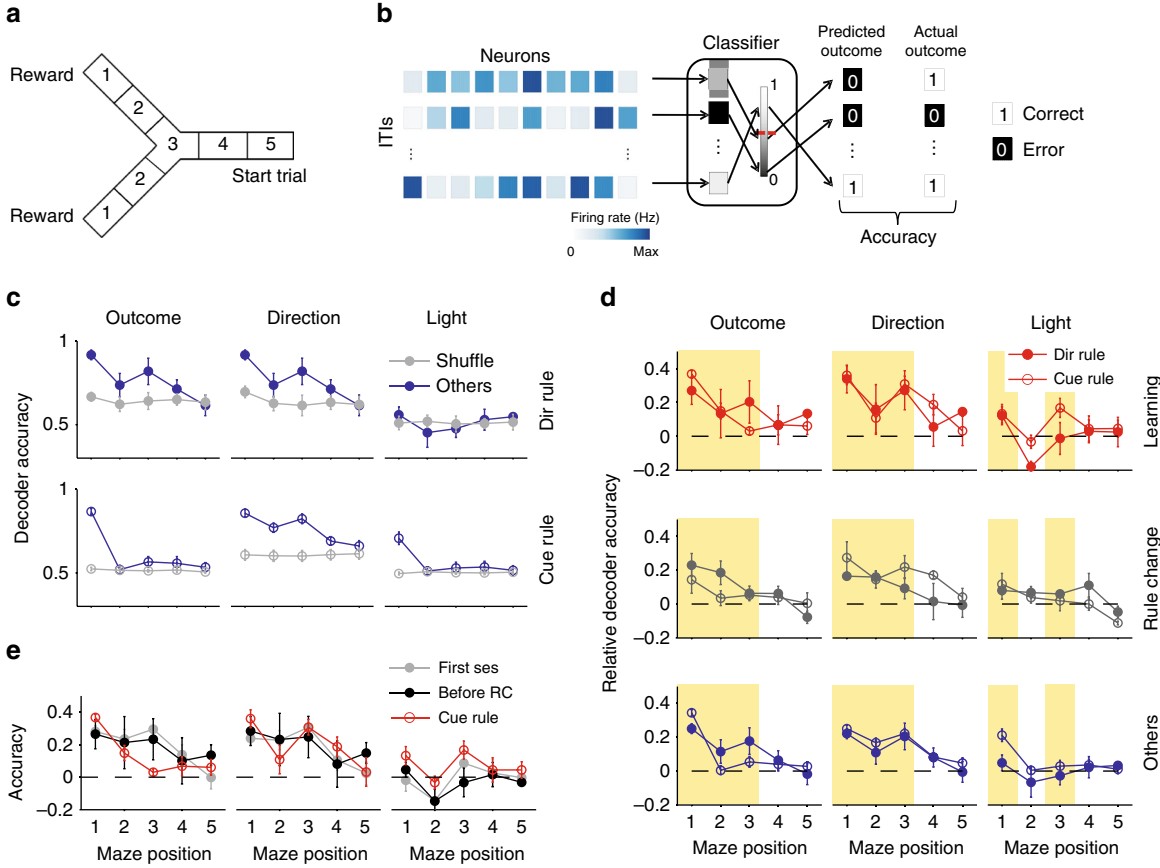

**Fig. 5** Position-dependent encoding of recent task-relevant information. **a** Division of the maze into five positions. Position 1 is the goal arm end. Position 3 is the choice point during the trial. **b** Schematic of decoding task events. For each inter-trial interval, and for each position in the maze, the population's firing rate vector is input to a linear decoder. The decoder is trained to classify the population vector with one of two possible labels (here, error or correct prior outcomes), given a threshold (red dashed line) on the decoder's output. The decoder's accuracy is the proportion of correctly predicted labels on held-out data. **c** Example decoder accuracy as a function of maze position for the "other" sessions. We plot absolute decoder performance for each of the three classified features (prior outcome, prior direction choice, and prior light positions), separated by the rule type (direction or light-cue rules). Each data point is the mean ± SEM accuracy over sessions. Chance performance is plotted in grey, defined separately for each session and rule type (Methods). All panels here plot results for a logistic regression decoder; other decoders are plotted in the Supplementary Fig. 9. **d** Relative decoder accuracy over all session types and rule types. Each data point is the mean ± SEM decoder accuracy in excess of chance (0; dashed line) over the indicated session type (learning, rule-change, other). Highlighted groups of positions indicate consistent departures from chance performance in at least one session type. We replot these results grouped by rule-type in Supplementary Fig. 8. **e** Decoding of task features at the start of learning. We plot the decoding accuracy over the first session of each animal (grey), over all sessions before the first rule change (black), and over the first light-cue session for each animal (red)

We tested this hypothesis by computing the similarity between the ensemble activity pattern in an inter-trial interval and the activity pattern of the same ensemble in the preceding trial. We found that the similarity between trial and inter-trial interval ensembles was indistinguishable from the similarity predicted between two sets of independently firing neurons (Supplementary Fig. 7a, b). This null result was robust across all session types, and irrespective of whether we grouped inter-trial intervals by trial outcome, choice of arm, or location of the light cue. Consistent with this, we found that the subset of neurons active on every trial and inter-trial interval did not show any recall effect during the inter-trial interval (Supplementary Fig. 7c, d). Thus the recalled patterns of ensemble activity in inter-trial intervals seem unrelated to the ensemble activity within the preceding trial.

**Mixed encoding of retrospective task information**. What then did the recalled activity pattern encode? Its transient appearance, immediately before behavioural change but fading before the end of a session, suggests a temporary representation, akin to short-term memory. That the recalled pattern was triggered only by

prior reinforcement suggests the hypothesis that the recalled ensemble was a working memory encoding of task features that were potentially relevant for learning. If it was a working memory for task features, then we should be able to decode prior task information from ensemble activity.

To address this, we assessed our ability to decode prior outcome, choice of goal arm, and light-cue position from the core population's activity. As prefrontal cortex activity encoding often shows broad position dependence[17, 20, 37], we divided the linearised maze into five equally spaced sections (Fig. 5a), and represented the core population's activity in each as the vector of its neurons' firing rates in that section. We used these firing rate vectors as inputs to a cross-validated linear decoder (Fig. 5b), and compared their predictive performance to shuffled data (Methods).

We could decode prior outcome, choice of goal arm direction, and cue position well above chance performance, and often in multiple contiguous maze positions. We plot the absolute decoding performance for the "other" sessions in Fig. 5c to illustrate that decoding at some maze positions was near-perfect, with some sessions decoded at 100% accuracy. The learning and

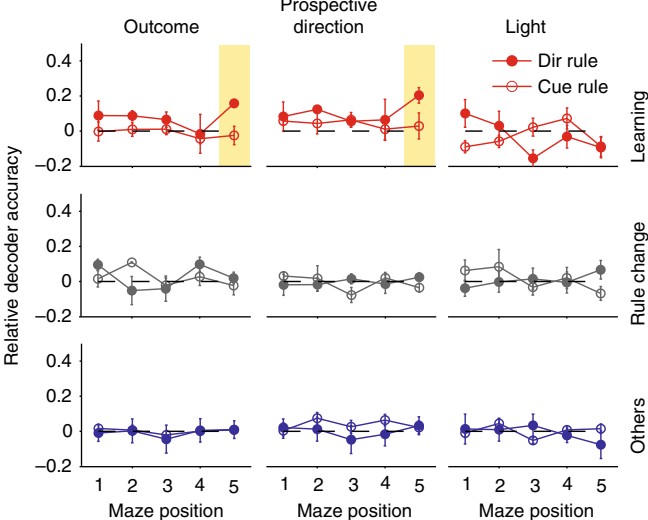

**Fig. 6** No prospective encoding of outcome and choice. We plot here the relative decoder accuracy over all session types and rule types for features on the immediately following trial. Compare with Fig. 5d. Each data point is the mean ± SEM accuracy in excess of chance (dashed line) over the indicated sessions (learning, rule-change, other). As a sanity check that our cross-validation of the decoder and shuffled controls were working, we also decoded the prospective light position: as this was randomised, the ensemble activity could not predict its position and so should only have been decoded at chance levels—which it was. We replot these results grouped by rule-type in Supplementary Fig. 8

rule-change sessions also had maze positions with near-perfect decoding across all sessions (Supplementary Fig. 8). These results were consistent across a range of linear decoders (Supplementary Fig. 9). Ensemble activity in medial prefrontal cortex thus robustly encoded multiple task events from the previous trial.

**Persistent and learnt encoding**. We then compared decoding performance between session and rule types. Chance decoding performance differed between task features (as the randomised light-cue was counter-balanced across trials, but each rat's choice and hence outcomes were not), and between session types and rule types (as rat performance differed between them). Thus we normalised each decoder's performance to its own control, and compared this relative decoding accuracy across sessions and rules (Fig. 5d).

These comparisons revealed we could decode the prior choice of direction (left or right) in all types of session and regardless of whether the rule was direction-based or cue-based (Fig. 5d). Decoding of direction choice was robustly above chance while the rats moved from the end of the goal arm back to the maze's choice point (highlighted yellow); on cued-rule sessions, this decoding extended almost all the way back to the start position of the departure arm. Accurate decoding of direction choice could be observed from the very first session of each rat, and consistently across sessions before the first rule change (Fig. 5e). These results indicated that medial prefrontal cortex always maintained a memory of prior choice, and did not need to learn to encode this task feature.

Similarly, we could decode the prior outcome (correct or error) in all types of session and regardless of whether the rule was direction- or cue-based (Fig. 5d). Decoding of outcome was notably stronger at the end of the goal arm, where the reward was delivered, but could also be decoded above chance while the rats traversed the maze back to the start position (highlighted yellow).

Nonetheless, decoding of outcome was again present from the very first session (Fig. 5e). These results indicated that medial prefrontal cortex always encoded the trial's outcome, and did not need to learn to encode this task feature.

By contrast to the encoding of prior direction and outcome, we could only reliably decode the prior cue position in two specific locations (Fig. 5d). The prior cue position was consistently encoded at the end of the goal arm for both cue and direction rules, likely corresponding to whether or not the light was on at the rat's position. But the only sustained encoding of prior cue position while the rat traversed the maze was during learning sessions for cue-based rules (yellow highlighted position and red open circles in Fig. 5d). There was no sustained encoding of the cue during learning sessions of direction rules (red filled circles in Fig. 5d). And this sustained encoding of the cue did not appear in the first session, nor in any session before the first change to the cue-based rule (Fig. 5e). Consequently, these results suggest that only in learning sessions did the core population encode the memory of the prior cue position, and only when relevant to the learnt rule.

When we examined single neuron tuning to preceding task features, we found only weak tuning in a handful of neurons (Supplementary Fig. 10), consistent with prior reports[8]. Unlike the population-level decoding, there was no difference in single neuron tuning between session types (Supplementary Fig. 10). Consequently, strong and differential retrospective encoding of task features appears only in the collective activity of the core population.

**No prospective encoding**. Strikingly, we found that decoding of prospective choice or outcome on the next trial was at chance levels throughout the inter-trial interval (Fig. 6). These results were consistent with our finding that the ensemble activity pattern preceding correct trials was not systematically recalled (Fig. 3c). They also show that the decoding of prior task features from the core populations' activity was non-trivial. The only above-chance decoding of prospective information was observed for direction-based rules, where we found that decoding of future choice and outcome was above chance level only for learning sessions and only at position 5, where the animal makes a U-turn before starting the new trial. This suggests that medial prefrontal cortex activity around the start of the trial could also be related to the upcoming decision when the task rule is successfully learnt.

## Discussion

We sought to understand how short-term memory in medial prefrontal cortex may support the trial-and-error learning of rules from a naive state. To do so, we analysed population activity in medial prefrontal cortex from rats learning rules on a maze, and asked if the activity during the inter-trial interval carried signatures of short-term memory for rule-relevant features of the task. We found that a specific pattern of ensemble activity was recalled only after reinforced trials, and only reliably during sessions in which the rats learnt the current rule for the first time. This dependence on prior outcome, and the transient appearance of the ensemble activity pattern, was consistent with a short-term memory encoding, rather than a persistent change to the underlying neural circuit.

We could robustly decode prior outcome and direction choice from ensemble activity across all sessions, but found that encoding of the prior cue position was specific to learning sessions for the cue-based rules. This suggests that the recalled ensemble is a repeated synchronisation of multiple encodings across the neural population, with rule-appropriate suppression or enhancement of cue encoding. We thus propose that

reinforcement tags features to sustain in medial prefrontal cortex working memory, and does this by reliably triggering a specific pattern of ensemble activity that jointly encodes relevant task features.

Our results support recent studies of prefrontal cortex population activity that reported how the pattern of population activity in rodent prefrontal cortex changes with or immediately prior to an internally-driven shift in behavioural strategy[10, 12]. We extend these prior results in three ways. First, prior work has studied scenarios where animals well-trained on one contingency experienced a change in that contingency. Here, we have shown that such abrupt shifts in population activity patterns can occur from the naive state. Consequently, they encode initial acquisition as well as uncertainty[11]. Second we have shown that such an abrupt shift in population activity happens for a putative working memory representation. Third, we have shown that this shift is selectively triggered by prior reinforcement. Nonetheless, our results add to the growing evidence that an abrupt shift in prefrontal cortex population activity is a necessary condition for the successful expression of a new behavioural strategy.

The observed shifts in prefrontal cortex population activity while both acquiring new strategies and switching between well-learnt strategies are potentially driven by distinct processes. In our task, we observed that the shift in population activity correlated with correct performance. Recent studies of strategy-switching in well-trained animals[2, 11, 12, 38] have reported that errors are the main trigger for the shift in population activity. Similarly, Narayanan et al.[39] report evidence that an error on a well-learnt task drives an increased coherence between medial prefrontal cortex and motor cortex on the subsequent trial, suggesting that medial prefrontal cortex plays a role in adapting performance after errors by top-down control. We propose these findings are respectively consistent with separate roles of the prefrontal cortex: that changes in its neural representations during the learning of novel environmental associations are driven by unexpected reward (as we report here), whereas changes in its neural representations during performance monitoring of learnt associations are driven by unexpected errors[2, 11, 12, 38, 39].

This hypothesis fits well with theories of reinforcement learning based on the firing of midbrain dopamine neurons[40–43]. In the naive state, unexpected reward elicits phasic dopamine activity; in the well-learnt state, reward is expected so elicits no dopamine response, but unexpected errors elicit phasic dips in dopamine activity. These bursts and dips can act as teaching signals for long-term plasticity[44] or modulatory signals for short-term changes in neural dynamics[35, 36]. As medial prefrontal cortex connects directly and via striatum to midbrain dopaminergic neurons[45] and receives direct input from them[46, 47], there is a potential direct link between reward-feedback mechanisms and adaptive behaviour[2].

That the recalled ensembles only appeared around clear episodes of behavioural learning means they are thus candidate cell assemblies[34]: an ensemble that appeared during the course of learning. We distinguished here between structural and functional cell assemblies. In a structural assembly, the ensemble's activity pattern is formed by some underlying physical change, such as synaptic plasticity of the connections between and into the neurons of the ensemble[34, 48, 49], and is thus a permanent change. In a functional assembly, the ensemble's activity pattern is formed by some temporary modulation of existing connections, by new input or neuromodulation[36], and is thus a temporary change. Our analysis suggested that the recalled ensembles were a functional assembly, as they decayed before the end of the session in which they appeared, often decaying soon after the learning trial itself. We thus propose that this short-term memory ensemble is necessary only for the successful trial-and-error

learning of a new rule, and not for the ongoing successful performance of that rule.

Consistent with prior reports of mixed selectivity in prefrontal cortex[6, 9, 24, 50], we could decode multiple task features from the joint activity of a population of neurons. Extending these reports, we showed here that these encodings were position dependent, and that this encoding was exclusively retrospective during the inter-trial interval—despite there being no explicit working memory component to the rules. Our data thus show a short-term memory for multiplexed task features even in the absence of overt working memory demands.

One of our more unexpected findings was that we could reliably decode both the prior choice of direction and the prior trial's outcome across all sessions, regardless of whether they contained clear learning, externally imposed rule changes, or neither these events. Our decoder used the vector of firing rates at a given maze position as input. Consequently, our ability to decode binary labels of prior events (correct/error trials or left/right locations) implies that there were well separated firing rate vectors for each of these labels. But this does not mean the neurons' firing rates were consistently related for a given label (such as a prior choice of the left arm of the maze). Indeed, it could imply anything from the two labels being encoded by the only two vectors of firing rates that ever appeared, to the two labels being encoded by two distinct groups of neurons whose firing rates within each group were never correlated. The reliable appearance of the same pattern of neuron co-activity only during learning thus implies that only during these sessions was the firing rate vector reliably similar. This suggests that learning to synchronise the encoded features, and not the learning of the encoding itself, is necessary for acquiring a new rule.

An interesting detail with potentially broad implications is that we could decode both the choice of prior direction and prior outcome from the very first session that each rat experienced the Y-maze. Either this implies that medial prefrontal cortex learnt representations of direction and outcome so fast that they were able to make a significant contribution to decoding by population activity within the very first session. Or it implies that medial prefrontal cortex does not need to learn representations of direction and outcome, meaning that such encoding is always present. Future work is needed to distinguish which of the broad spectrum of features encoded by the prefrontal cortex are either consistently present or learnt according to task demands. Demarcating the classes of features that the prefrontal cortex innately or learns to remember would further advance our understanding of its contribution to adaptive behaviour.

## Methods

**Task description and electrophysiological data.** For full details on training, spike-sorting, and histology see ref. [26]. All experiments in that study were carried out in accordance with institutional (CNRS Comité Opérationnel pour l'Ethique dans les Sciences de la Vie) and international (US National Institute of Health guidelines) standards and legal regulations (Certificate no. 7186, French Ministère de l'Agriculture et de la Pêche) regarding the use and care of animals.

Four Long-Evans male rats with implanted tetrodes in prelimbic cortex were trained on a Y-maze task (Fig. 1a). Each recording session consisted of a 20–30 min sleep or rest epoch, in which the rat remained undisturbed in a padded flowerpot placed on the central platform of the maze, followed by a training epoch, in which the rat performed for 20–40 min, and then by a second 20–30 min sleep or rest epoch. Every sleep epoch contained periods of slow-wave sleep, which were detected offline automatically from local field potential recordings (details in ref.[26]).

The Y-maze had symmetrical arms, 85 cm long, 8 cm wide, and separated by 120 degrees, connected to a central circular platform (denoted as the choice point throughout). During training, every trial started when the rat left the beginning of the start arm and finished when the rat reached the reward port at the end of its chosen goal arm. A correct choice of arm according to the current rule was rewarded with drops of flavoured milk. Each inter-trial interval lasted from the end-point of the trial, though any reward consumption, until the rat completed its self-paced return to the beginning of the start arm.

Each rat had to learn the current rule by trial-and-error. The rules were sequenced to ensure cross-modal shifts: go to the right arm; go to the cued arm; go to the left arm; go to the uncued arm. To maintain consistent context across all sessions, the light cues were lit in a pseudo-random sequence across trials, whether they were relevant to the rule or not.

The data analysed here were from a total set of 53 experimental sessions taken from the study of Peyrache et al.[26], representing a set of training sessions from naive until either the final training session, or until choice became habitual across multiple consecutive sessions (consistent selection of one arm that was not the correct arm). In this data-set, each rat learnt at least two rules, and the four rats respectively contributed 14, 14, 11 and 14 sessions. The learning, rule change, and "other" sessions for each rat were intermingled. We used 50 sessions here, omitting one session for missing position data, one in which the rat always chose the right arm (in a dark arm rule) preventing further decoding analyses (see below), and one for missing spike data in a few trials. Tetrode recordings were obtained from the first session for each rat. They were spike-sorted only within each recording session for conservative identification of stable single units. In the sessions we analyse here, the populations ranged in size from 15–55 units. Spikes were recorded with a resolution of 0.1 ms. Simultaneous tracking of the rat's position was recorded at 30 Hz.

**Behavioural analysis.** A learning trial was defined following the criteria of the original study[26] as the first of three correct trials after which the performance was at least 80% correct for the remainder of a session. Only ten sessions contained a trial which met these criteria, and so were labelled "learning" sessions. We checked that these identified trials corresponded to an abrupt change in behaviour by computing the cumulative reward curve, then fitting a piecewise linear regression model: a robust regression line fitted to the reward curve before the learning trial, and another fitted to the reward curve after the learning trial. The slopes of the two lines thus gave us the rate of reward accumulation before ($r_{before}$) and after ($r_{after}$) the learning trial.

To identify other possible learning trials within each learning session, we fitted this piecewise linear regression model to each trial in turn (allowing a minimum of five trials before and after each tested trial). We then found the trial at which the increase in slope ($r_{after} - r_{before}$) was maximised, indicating the point of steepest inflection in the cumulative reward curve. The two sets of learning trials largely agreed: we checked our results using this set too.

Amongst the other sessions, we searched for signs of incremental learning by again fitting the piecewise linear regression model to each trial in turn, and looking for any trial for which ($r_{after} - r_{before}$) was positive. We found 22 sessions falling in this category in addition to the 10 learning sessions. We called those new sessions "minor-learning" (Supplementary Fig. 3).

**Testing for reinforcement-driven ensembles.** In order to identify ensembles and track them over each session, we first selected the $N$ neurons that were active in all the inter-trial intervals. The $N$ spike trains of this core population were convolved with a Gaussian ($\sigma = 100$ ms) to obtain a spike-density function $f_k$ for the $k$th spike train. All the recall analysis was repeated for different Gaussian widths ranging from 20 to 240 ms (Supplementary Fig. 5). Each spike train was then $z$-scored to obtain a normalised spike-density function $f^*$ of unit variance: $f_k^* = (f_k - \langle f_k \rangle)/\sigma_k$, where $\langle f_k \rangle$ is the mean of $f_k$, and $\sigma_k$ its standard deviation, taken over all the inter-trial intervals of a session.

We sought to compare the co-activity of neurons within the core population across the inter-trial intervals of a session, in order to determine if the same pattern of co-activity recurred. To do so, for each inter-trial interval $t$ we first computed a pairwise similarity matrix $\mathbf{S_t}$ between the spike-density functions for all $N$ neurons. Similarity here was the rectified correlation coefficient, retaining all positive values, and setting all negative values to zero, such that each entry $S_t(i, j)$ for the pair of neurons $(i, j)$ was in the range [0, 1], from 0 meaning never co-active to 1 meaning identically co-active.

We then compared the core population's co-activity patterns between inter-trial intervals $t$ and $u$ by computing the pairwise similarity between $\mathbf{S_t}$ and $\mathbf{S_u}$. We do this by computing the rectified correlation coefficient between the vectors of all values above the diagonal in $\mathbf{S_t}$ and $\mathbf{S_u}$, giving a scalar value $R(t, u) \in [0, 1]$. Note that this is why negative correlations between neurons were omitted: if we had not, then positive $R(t, u)$ could correspond to either a set of neurons that were similarly correlated in both intervals, or a set of neurons that were similarly anti-correlated in both intervals. Thus we used only pairwise similarity between neurons to disambiguate these two cases, and specifically identify co-activity between neurons.

By computing $R(t, u)$ for each pair of inter-trial intervals in a session, we thus formed the recall matrix $\mathbf{R}$, capturing the similarity of activity patterns between all inter-trial intervals in that session. We grouped the entries of $\mathbf{R}$ into two groups according to the same type of inter-trial interval–predominantly whether they were intervals following correct or following error trials. These created the block diagonals $\mathbf{R_1}$ and $\mathbf{R_2}$ (such as $\mathbf{R_{error}}$ and $\mathbf{R_{correct}}$, as illustrated in Fig. 2c). We summarised the recall between groups by computing the mean of each block. We detected statistically meaningful differences by computing the Kolmogorov–Smirnov test for a difference between the distributions of values in the two blocks.

In the main text, we report that there is higher average similarity in $\mathbf{R_{correct}}$ than $\mathbf{R_{error}}$ in many sessions. However, there was a strong tendency for inter-trial intervals following correct trials to be longer in duration than inter-trial intervals following error trials (Supplementary Fig. 2), and so the estimates of pairwise similarity may be biased. In order to dissect the contribution of the different durations we defined a null model for the expected similarity between intervals due to their durations alone. For each session we created a predicted recall matrix $\hat{\mathbf{R}}$, by averaging 1000 random recall matrices, each computed from shuffled spike trains. Each spike train was shuffled by randomly re-ordering its inter-spike intervals. In this way we destroyed any task-specific temporal pattern of the spike train, thus quantifying the contribution to its pairwise similarity with other neurons solely due to the duration of the inter-trial interval. Our final residual recall matrix $\tilde{\mathbf{R}} = \mathbf{R} - \hat{\mathbf{R}}$ was obtained as the difference between the Recall matrix and the average shuffled recall matrix (Fig. 2c; Supplementary Fig. 3).

This dissection of the contribution of duration was why we used similarity ([0, 1]) rather than correlation ([−1, 1]) between $\mathbf{S_t}$ and $\mathbf{S_u}$ to compute $R(t, u)$, and consequently $\hat{R}(t, u)$. First because it then allowed the residual recall value $\tilde{R}(t, u)$ to fall in the range [−1, 1]. Second because, if we had used correlation between $\mathbf{S_t}$ and $\mathbf{S_u}$, then if both $R(t, u)$, and $\hat{R}(t, u)$ were negative and $R(t, u) > \hat{R}(t, u)$, then the residual recall value would be positive ($\tilde{R}(t, u) > 0$), thus indicating a similarity between co-activity patterns in intervals $t$ and $u$ despite both data and shuffled recall matrices indicating a dissimilarity.

For the residual recall matrix, we summarised and tested the differences between the two groups (such as post-error and post-correct inter-trial intervals) in the same way as detailed above, given the new block diagonals $\tilde{\mathbf{R}}_1$ and $\tilde{\mathbf{R}}_2$. When grouping by session type, we plot the difference between the block diagonals' means as $\Delta_R = \text{mean}(\tilde{\mathbf{R}}_1) - \text{mean}(\tilde{\mathbf{R}}_2)$.

To test the likelihood of obtaining ten sessions with greater recall for post-correct inter-trial intervals by chance, we used a permutation test against the null model that the difference in recall ($\Delta_R$) for outcomes was randomly distributed across the sessions. We repeatedly chose ten sessions at random without replacement from the 50, and observed if those ten all had positive $\Delta_R$; we repeated this 10,000 times.

For the path length analysis, we computed the distance travelled by the rat from the start to the end of each inter-trial interval from the vectors given by the frame-by-frame $(x, y)$ co-ordinates. For a recall analysis based on the path lengths, we divided the inter-trial intervals of a session into groups of "short" or "long" paths, based on whether the path length in that interval was below or above the median path length for the session. We then grouped the session's residual recall matrix by these two groups, forming the two blocks $\mathbf{R_{short}}$ and $\mathbf{R_{long}}$, and performed the same recall analysis as outlined above on the differences between the two blocks. (We did the same analysis using a k-means clustering of the path lengths in a session into two groups; the group sizes were often highly asymmetric, with few intervals in the "long" group, making the comparison of recall between short and long groups biased. Nonetheless, the results for the differences in recall between short and long groups were qualitatively the same for all session types).

We tested whether the recalled pattern of ensemble activity in the inter-trial interval was a replay of the pattern seen in the preceding trial. To do this, we used the same approach as the recall analysis between inter-trial intervals: we computed pairwise neuron similarity matrices for the inter-trial interval $\mathbf{S_{ITI}}$ and the preceding trial $\mathbf{S_{trial}}$, and computed the similarity $R_{within}$ between those matrices. For consistency with the other analyses, we created a core population of neurons that were active on all trials, and formed the similarity matrices from those neurons in both the trial and the inter-trial interval. As trials were typically 4 s long, this reduced the number of neurons in the core population compared to the full set used for just the comparisons between inter-trial intervals. We defined a null model for the predicted similarity between independently firing groups of neurons by shuffling inter-trial interval spike trains, computing the similarity matrix $\mathbf{S^*_{ITI}}$ for these shuffled spike trains, and computing the similarity $R^*_{within}$ between $\mathbf{S_{trial}}$ and $\mathbf{S^*_{ITI}}$. The difference between the data similarity and the null model's similarity was $\Delta_{within} = R_{within} - R^*_{within}$; $\Delta_{within} > 0$ would thus indicate that the ensemble activity patterns in the inter-trial interval and preceding trial were more similar than predicted by independent patterns. We repeated the shuffling 50 times. For each session we computed a one-tailed sign-test that $\Delta_{within}$ was greater than zero.

**Testing the onset and offset of recall.** In order to identify when the recalled ensemble activity pattern first appeared in a learning session, we arranged its residual recall matrix in trial order. For each trial in turn (with a minimum of three trials before and five after), we formed the block diagonals $\mathbf{R_{before}}$ and $\mathbf{R_{after}}$ (see Fig. 4a), respectively giving all pairwise recall scores between inter-trial intervals before and after that trial. The distance between recall before and after that trial was measured using the Kolmogorov–Smirnov statistic: the maximum distance between the empirical cumulative distributions of $\mathbf{R_{before}}$ and $\mathbf{R_{after}}$. The trial that had the maximum positive distance (an increase in recall from $\mathbf{R_{before}}$ to $\mathbf{R_{after}}$) and had $P < 0.05$ was identified as the onset of the recalled activity pattern. Similarly, the trial with the maximum negative distance that corresponded to a decrease in recall from $\mathbf{R_{before}}$ to $\mathbf{R_{after}}$ and had $P < 0.05$ was identified as the offset of the recalled activity pattern. In all learning sessions we observed a strict ordering of onset occurring before offset, and both occurring before the final tested trial of the session.

**Decoder analysis.** To test whether it was possible to predict task-relevant information in a position-dependent manner from the core population's activity we trained and tested a range of linear decoders[51]. In the main text we report the results obtained using a logistic regression classifier, as this is perhaps the easiest classifier to interpret.

We first linearised the maze in five equally sized sections, with the central section covering the choice point of the maze. During each inter-trial interval, we computed the $N$-length firing rate vector $\mathbf{r}^p$, with each element $r_j^p$ being the firing rate of the $j$th core population neuron at position $p$. For each session of $T$ inter-trial intervals and each section of the maze $p$, the set of population firing rate vectors $\{\mathbf{r}^p(1), …, \mathbf{r}^p(T)\}$ was then used to train a linear decoder to classify the relevant binary task information, either: the previous trial's outcome (labels: 0, 1), the previously chosen arm (labels: left, right), or the previous position of the light cue (labels: left, right). (We also trained all decoders on the next outcome, arm choice, and light position to test for prospective encoding). To avoid overfitting, we used leave-one-out cross-validation, where each inter-trial interval was held out in turn as the test target and the decoder was trained on the $T − 1$ remaining inter-trial intervals. The accuracy of the decoder for position $p$ in a given session was thus the proportion of correctly predicted labels over the $T$ held out inter-trial intervals.

Because the frequency of outcomes and arm choices were due to the rat's behaviour, chance proportions of correctly decoding labels was not 50%. To establish chance performance for each decoding, we fitted the same cross-validated classifier on the same set of firing rate vectors at each position, but using shuffled labels across the inter-trial intervals (for example, we shuffled the outcomes of the previous trial randomly). We repeated the shuffling and fitting 50 times. For displaying the results in Fig. 5, we subtracted the mean of the shuffled results from the true decoding performance. Separate results for the true and shuffled decoders are plotted in Supplementary Fig. 8a.

We report in the main text the results of using a logistic regression classifier. To check the robustness of our results, we also tested three further linear decoders: linear discriminant analysis; (linear) support vector machines; and a nearest neighbours classifier. Each of these showed similar decoding performance to the logistic regression classifier (Supplementary Fig. 9).

A single neuron's tuning at each maze position was assessed by splitting its firing rates at that position into two groups according to the preceding task feature (reward or error; left or right direction choice; left or right cue), then applying a Kolmogorov–Smirnov test for a difference between the two groups: tuned neurons were those with $P < 0.05$.

**Data availability.** The spike train and behavioural data that support the findings of this study are available in CRCNS.org (https://doi.org/10.6080/K0KH0KH5) (ref. [52]). Code to reproduce the main results of the paper is available at: https://github.com/sibangi/PFCensemble_ITI.

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

## Acknowledgements

We thank Matt Jones for comments on the manuscript, and the Humphries' lab (Abhinav Singh, Javier Caballero, Mat Evans) for discussion. M.D.H. and S.M. were supported by a Medical Research Council Senior non-Clinical Fellowship award [MR/J008648/1] and Medical Research Council Project Grant [MR/P005659/1]. The original data collection was supported by the EU Framework (FP6) "ICEA" grant. A.P. was supported by a Canada Research Chair Tier 2 (154808).

## Author contributions

M.D.H. and S.M. designed the analyses. S.M. analysed the data. A.P. provided data and advice on the experiments. M.D.H. and S.M. wrote the manuscript, with assistance from A.P.

## Additional information

**Competing interests:** The authors declare no competing interests.

