## [Peer Review File · Nature Communications]

Reviewers' comments:

Reviewer #1 (Remarks to the Author):

The study by Maggi and colleagues reports interesting new findings from a previously published data set (Peyrache et al., 2009). They report evidence for transient changes in the coordinated activity of groups of neurons from the rat medial prefrontal cortex as the rats experienced changes in reward locations in a Y-maze task. The main finding is that retrospective spatial information is encoded by ensemble activity prominently during the learning phase of each block of trials and in an outcome-dependent manner. This aspect of the study is the main novel finding of the paper.

Three issues were apparent to me when reading the paper and I hope that the authors may have some data to address these issues.

First, as LFPs were recorded according to the methods, was there a relationship LFP power (in theta or gamma) that might account for the transient nature of the ensemble activity after correct and error trials? This would help link their findings to the Laubach and Narayanan papers that were cited. Post-error theta was a major finding from that group, especially their 2013 paper in Nature Neuroscience (which was not mentioned by Maggi and colleagues).

Second, early papers on memory-related activity by the rodent mPFC reported weak mnemonic signals at the single cell level (Jung et al. 1998). Later studies, by some of the same researchers, report clear ensemble decodings of mnemonic information by mPFC ensembles (Baeg et al., 2003). Can the authors provide any data on how their single units decode spatial information as a function of outcome? I did not find a summary of single unit activity anywhere in the manuscript.

Third, as position was recorded and used for several of the analyses, what is the outcome of using position alone, and not neural activity, as the input variable for the same data analyses that are reported in the paper? To what extent would position by itself account for retrospective recall? I raise this issue due to several reports of behavioral variability having a major role in driving mPFC activity, papers by Euston and Cowen with McNaughton, which were not mentioned by Maggi and colleagues.

Reviewer #2 (Remarks to the Author):

Maggi et al, record from mPFC during a y-maze task, and report that neuronal ensembles in 4 rats encode retrospective aspects of reinforcement. This ensemble code seemed to change as animals learned the task, being particularly prominent around a transition in learning. The ensembles also seemed to decode many aspects of task performance. I thought it was well-done, well-written and interesting, and I thought the changes with learning were particularly intriguing, but I had trouble understanding the details of analyses and I'm concerned about motor control during the ITI.

I have some questions that might help me understand the manuscript better:

1) Based on the paper, I have a poor intuition, based on the data presented, of what the neurons are doing. Figure 1b is key - this is after a correct trial? - I might show what happens after an error to neuronal firing rate, and show the same ensemble at different stages in learning. At a rate, there may be too many neurons to really see the pattern - picking three neurons, and showing the rasters for correct vs error early vs. late in learning would be really helpful.

2) I also am having trouble understanding the recall metric. I would slowly unpack this for the reader

– perhaps expanding Figure 2. I understand that there is a quantity S_t which is the correlation between spike density functions of two neurons, but on line 497, what is S_u ? I might take the approach of showing two neurons, showing what S_t looks like, then showing S_u , and then showing R_t for that ensemble. By the time I get to Figure 2, I can't understand it the way it's currently presented.

3) Do authors track movement during the ITI? The line in Figure 1a seems to be drawn by hand. One could argue that rats are doing wildly different things during the ITI on correct vs. error trials and during learning, and that's driving the effect.

4) Position tracking data might also help understand Figure 5. Are the time bins equal for each zone?

5) How is 'residual recall' calculated? What does it mean? On line 141, it says that this is recall subtracted from 'common duration'

6) What is the z-axis on Figure 2a? Recall?

7) What is meant by the 'core' population - only neurons active in every intertrial interval? There may be neurons that come on for satiety, with changes in learning rate, etc. What happens if all neurons are included?

Minor:

Line 36: This hypothesis is quite complex

Line 79: Behavioral history prior to implants might be helpful

The use of boxplots / single rat data are admirable, but obscure the main effects - for instance, are the boxes in Figure 2c same or different?

Reviewer #3 (Remarks to the Author):

Maggi and colleagues investigated the neuronal ensemble activity patterns during inter-trial intervals (ITIs) in a rule-learning task. They quantified pairwise correlations between firing activities of single neurons during ITIs, and compared similarities of them across different ITIs. They found that similarity of ensemble co-activity patterns during ITI increased after the animals learned the rule of the task. Furthermore, they were able to decode the information of the task events from the ensemble activity patterns. The manuscript is very clear and well written. The topic is one of great interest and it will provide large impacts for a wide audience in the fields.

I have one major comment. In this study, the authors quantified only the neural activities during ITI periods in most of the analysis. Similarity of ensemble co-activity patterns of was assessed across different ITI periods. I think that comparisons of the ensemble activity patterns between the task trials and ITIs are also important. The analysis will provide the information whether the ensemble activity during ITI was recall of the activity in trail periods (e.g., Peyrache et al., 2009), or, it was different from the activity in task trials and represented some kind of context information (e.g. Farovik et al., 2015).

Minor points:

1) Line #136: "To control for this, we used shuffled spike-trains to compute the expected pairwise similarities between neurons due to just the duration of each interval..."

How the authors shuffled spike-trains is not clear. Please explain it here or in Methods.

2) Line #495: About the method to obtain a normalized spike-density function. Which periods did the authors use for calculating mean $\langle f_k \rangle$ and standard deviation σ_k ? Whole of a session, or only ITI periods?

Response to reviewers

We thank the reviewers for their positive comments and constructive suggestions. The majority of the revision is in reporting the results of new analyses suggested by the reviewers: these are in three new and one extended Supplemental figure, along with corresponding descriptions in the main text..

The data set underlying the analyses in this paper is in the process of being made publicly available on CRCNS.org. We will add the DOI to final version of this manuscript.

Reviewer #1 (Remarks to the Author):

The study by Maggi and colleagues reports interesting new findings from a previously published data set (Peyrache et al., 2009). They report evidence for transient changes in the coordinated activity of groups of neurons from the rat medial prefrontal cortex as the rats experienced changes in reward locations in a Y-maze task. The main finding is that retrospective spatial information is encoded by ensemble activity prominently during the learning phase of each block of trials and in an outcome-dependent manner. This aspect of the study is the main novel finding of the paper.

Three issues were apparent to me when reading the paper and I hope that the authors may have some data to address these issues.

First, as LFPs were recorded according to the methods, was there a relationship LFP power (in theta or gamma) that might account for the transient nature of the ensemble activity after correct and error trials? This would help link their findings to the Laubach and Narayanan papers that were cited. Post-error theta was a major finding from that group, especially their 2013 paper in Nature Neuroscience (which was not mentioned by Maggi and colleagues).

The referee is quite right that we should have included reference to the Narayanan et al 2013 Nature Neurosci paper in the discussion of our proposed hypotheses for separate roles of prefrontal cortex in adapting performance in learning and in monitoring of well-learned behaviours. We have now added explicit discussion of that paper to our Discussion on lines 423-426, and rewritten the paragraph (lines 420-432) to more clearly place that work in the context of our hypothesised roles in adaptive behaviour.

However, as we note there, the post-error theta in PFC LFPs reported by Narayanan et al 2013 is a signature of performance monitoring, whereas our aim is to study the recruitment of, use, and changes to short-term memory during learning. Specifically, they report a brief increase in theta-band power in their PFC LFPs in a trial that follows an error trial. As this is in well-trained animals, and was shown to modulate motor cortex LFPs, Narayanan et al interpret their results as a sign of top-down control of movement to correct an error. Moreover, as we noted in the Discussion, our results suggest a clear dichotomy of which feedback is important: correct outcomes during initial learning of action-outcome associations, and errors for performance monitoring. Thus we believe that the “post-error theta” findings, while complementary, make no predictions for what we might expect from

LFP activity during learning, nor what we would expect to find immediately after an outcome. Consequently, in the absence of any predictions, we have not analysed LFP activity here as it would be an open-ended exploratory analysis unrelated to our main aims.

Second, early papers on memory-related activity by the rodent mPFC reported weak mnemonic signals at the single cell level (Jung et al. 1998). Later studies, by some of the same researchers, report clear ensemble decodings of mnemonic information by mPFC ensembles (Baeg et al., 2003). Can the authors provide any data on how their single units decode spatial information as a function of outcome? I did not find a summary of single unit activity anywhere in the manuscript.

We agree that this is a necessary analysis in order to interpret the population decoding: for if the single units also readily encode prior task features then the population decoding is a straightforward consequence of their output, and not a feature specific to the population's joint activity. We had in fact done a preliminary analysis of single neuron coding in order to make sure that the population decoding was not a simple reflection of their output. Consequently, we have now analysed the dependence of each neuron's firing rate on either the outcome, direction or light cue, and at the same set of locations as the population decoder. We observed that:

- The number of neurons that significantly modulated their firing rate as function of outcome, direction and light is very small, consistent with Jung et al's weak encoding.
- The overall number of single units with a significant modulation of their rate does not differ between session types. Together these results show that indeed we gain more information on the mPFC's coding of task features from the population than from single units.

In addition, we found that the few modulated neurons encoded retrospective and not prospective information, and that the number of modulated neurons decreases along the maze when moving from the reward location back to the start position. Both these results replicate our findings using the linear decoders on the population's joint activity.

We added those results in the new version of the manuscript (see Supplementary Figure S10 and lines 354-359 in the main text).

Third, as position was recorded and used for several of the analyses, what is the outcome of using position alone, and not neural activity, as the input variable for the same data analyses that are reported in the paper? To what extent would position by itself account for retrospective recall? I raise this issue due to several reports of behavioral variability having a major role in driving mPFC activity, papers by Euston and Cowen with McNaughton, which were not mentioned by Maggi and colleagues.

Indeed mPFC activity has been linked with variations in behaviour that are independent of the task demands (Euston and McNaughton, 2006; Fujisawa et al., 2008). Aware of these properties of mPFC we had already taken steps in the manuscript in order to rule out sources of behavioural variability that could contribute to the Recall effect. In particular we controlled for:

- Different amount of time spent along the maze between error and correct intervals (Supplementary Figure S2, in the new version of the manuscript)
- Different trajectories, in terms of whether the rat was coming from the right or left arm of the maze. We explicitly showed that there was no recall effect based on arm direction (Figure 3).
- Difference in reward consumption following correct respect to error trials. We showed that there was no reward-location specific recall of ensemble activity (Supplementary Figure S4).

However, we agree with the reviewer that we had not directly addressed the potential issue of trajectory variations identified by Euston and McNaughton (2006). Unlike Euston and McNaughton's open-field, the narrow arms of the Y-maze (8 cm in width) prevent the wide variation in curved trajectories that underpinned their qualitative differences in mPFC single neuron activity. However, in that spirit, in our data there are clear variations in the path lengths taken from the reward position to the start position on the Y-maze, ranging from short, direct paths to long, explorative paths: such behavioural variations in path length could potentially give rise to different mPFC activity.

We quantified the path lengths and their relationship to the "recall" effect, and found no consistent difference in path length between post-correct and post-error inter-trial intervals, including when considering Learning sessions alone. Consequently, our finding of systematic recall of ensemble activity in intervals following correct outcomes could not be explained by a consistent difference in path length (Supplementary Figure S6). We also checked whether the path length itself drove a recall effect, by assessing whether the pattern of ensemble activity was more similar between intervals with short path lengths than between intervals with long path lengths, or vice-versa. Here we found no path-length dependent recall in Learning sessions, again ruling out path length as an explanation for our reinforcement-driven recall. However, when considering all 50 sessions together, we found that the short path-lengths tended to weakly drive more similar ensemble activity - we report this interesting finding too, as it provides another example of behavioural variation influencing mPFC activity.

We added a Supplementary Figure (S6) and we explained the new findings in the main text (lines 191-212).

Reviewer #2 (Remarks to the Author):

Maggi et al, record from mPFC during a y-maze task, and report that neuronal ensembles in 4 rats encode retrospective aspects of reinforcement. This ensemble code seemed to change as animals learned the task, being particularly prominent around a transition in learning. The ensembles also seemed to decode many aspects of task performance. I thought it was well-done, well-written and interesting, and I thought the changes with learning were particularly intriguing, but I had trouble understanding the details of analyses and I'm concerned about motor control during the ITI.

We thank the reviewer for the positive feedback regarding our paper. We revised the manuscript adding new analyses and changing some figures to make our methods and claims more clear and robust.

I have some questions that might help me understand the manuscript better:

1) Based on the paper, I have a poor intuition, based on the data presented, of what the neurons are doing. Figure 1b is key - this is after a correct trial? - I might show what happens after an error to neuronal firing rate, and show the same ensemble at different stages in learning. At a rate, there may be too many neurons to really see the pattern - picking three neurons, and showing the rasters for correct vs error early vs. late in learning would be really helpful.

In this new version of the manuscript, we modified Figure 1b, showing 4 examples of medial prefrontal cortex population activity within the same session (a learning session). In particular we show the heatmaps of convolved spike-trains of two ITIs following error trials (top panels) and two ITIs following correct trials (bottom panels). After a close look at the four panels it emerged that the two correct ITIs showed more similar pattern of spiking, in particular some neurons during correct ITIs increased their firing at the beginning of the recording and other towards the end compared to a more widely distributed firing of the same neurons during error ITIs.

We hope that this new version of the Figure 1, combined with the changes introduced also in Figure 2, will help the reader to better follow the manuscript.

2) I also am having trouble understanding the recall metric. I would slowly unpack this for the reader – perhaps expanding Figure 2. I understand that there is a quantity S_t which is the correlation between spike density functions of two neurons, but on line 497, what is S_u ? I might take the approach of showing two neurons, showing what S_t looks like, then showing S_u , and then showing R_t for that ensemble. By the time I get to Figure 2, I can't understand it the way it's currently presented.

We understand that the procedure used to quantify the recall effect might be complicated to follow. We decided therefore, as suggested by the reviewer, to unpack the methods and show the intermediate steps.

We modified Figure 2, introducing four example Similarity matrices. In particular, we computed the pairwise similarity matrix for each of the ITI shown in Figure 1b. In Figure 2a appeared very clearly that the two ITIs following correct trials had a more similar pattern of activity than the two ITIs following error trials. In order to further quantify how similar the activity pattern after correct was compared to the error trials we computed the similarity between the matrices shown in panel a. In Figure 2b we can observe that the correct ITIs have a more similar pattern (Recall) of ensemble activity compared to the errors. If we repeat the same procedure for multiple ITIs we can obtain our “recall” matrix as the pairwise similarity between all the similarity matrices (as shown in Figure 2c).

3) Do authors track movement during the ITI? The line in Figure 1a seems to be drawn by hand. One could argue that rats are doing wildly different things during the ITI on correct vs. error trials and during learning, and that's driving the effect.

Indeed Figure 1a is a schematic representation of an ITI path, and is not reflecting what the animal is doing. We agree with the reviewer that rats are probably covering the maze with different trajectories and this might contribute to the effect we observed. In order to tease apart whether the recall effect was due to an outcome-dependent activity or other behavioural variability we decided to investigate whether the animals were more or less explorative along the maze.

Considering that the Y-maze had very narrow arms (8 cm wide) we looked at the variations in the length of the path back to the start. We observed that the path length between post-error and post-correct trials was comparable for all session type (Supplementary Figure S6a in the new version of the manuscript). A major difference in the path length was observed at the reward location but not in the rest of the maze (Supplementary Figure S6b,c). Those results combined with our previous result (Supplementary Figure S4) exclude a possible path length contribution to the recall effect. Moreover we investigated whether differences in path length along the maze (excluding the reward location) could account for the reinforcement-related result. Again we couldn't observe a consistent difference in recall between short and long paths, ruling out path length as an explanatory variable for reinforcement-driven recall (Supplementary Figure S6d,e).

We thank the reviewer for this comment. We added a supplementary figure (S6) which is explained in the main text at lines 191-212.

4) Position tracking data might also help understand Figure 5. Are the time bins equal for each zone?

Supplementary Figure S2b shows that the time bins were not the same for each zone. Our aim for the decoder was to assess any position dependence of encoding the retrospective event, as shown in Figure 5. The decoder analysis took as input the core population's activity in each location as the vector of its neuron's firing rates (spikes/s), thus normalising the spike rate by time spent.

5) . How is 'residual recall' calculated? What does it mean? On line 141, it says that this is recall subtracted from 'common duration'

This was indeed unclear. We now write in the main text at line 136: "Consequently, by subtracting this expected recall matrix from the data-derived recall matrix, we obtained a ``residual" recall matrix: the similarity between ensemble activity patterns that remained after any effect of the durations of the inter-trials had been factored out (Figure 2c)". We have also redrafted the steps to construct the residual recall matrix on lines 596-600 of the Methods.

6) What is the z-axis on Figure 2a? Recall?

We modified Figure 2a, which is now Figure 2c accordingly. We inserted the legend for the z-axis.

7) What is meant by the 'core' population - only neurons active in every intertrial interval? There may be neurons that come on for satiety, with changes in learning rate, etc. What happens if all neurons are included?

In our analysis, we considered as the 'core' population only the neurons active in every intertrial interval. This choice has been partially imposed by the methods we chose. In particular, the recall matrix is meant to track the changes of network dynamics on a ITI-by-ITI basis, therefore we needed to track the same neurons along the entire session. However, as the reviewer highlighted, the excluded neurons might revealed learning-related activation or inactivation. We checked for this possibility and we observed that the proportion of ITIs in which each excluded neurons was active was unrelated to learning, or indeed externally imposed rule-changes. We concluded therefore that these neurons not in the core population were not necessarily relevant for learning.

We added those results in the Supplementary Figure S1 and described in the main text (lines 103-105).

Minor:

Line 36: This hypothesis is quite complex

We have rephrased this sentence (now line 34) to make clear the hypothesised mechanism, and its consequence: "An hypothesis we consider here is that reinforcement tags preceding choices and features to remember, in order to learn the rules of the environment." We have also removed the final sentence of this paragraph (which said "We thus sought to test the hypothesis that ensemble activity in medial prefrontal cortex represents a short-term memory of task features and choices that are potentially necessary for learning the current rule.") - this complex idea is laid out more clearly in the existing subsequent paragraph, making this sentence redundant.

Line 79: Behavioral history prior to implants might be helpful

As we noted in the manuscript, there was no behavioral history prior to implanting the tetrodes - the first session with implants was the first session in the maze.

The use of boxplots / single rat data are admirable, but obscure the main effects - for instance, are the boxes in Figure 2c same or different?

With Figure 2c (now Figure 2e) we wanted to see if the ensemble activity showed a more similar pattern during correct than during error ITIs in any session type. We plot this for each session as the difference between the mean recall for correct and the mean recall for error trials. Thus the key test is whether this difference in recall is greater than zero; i.e. within session type, rather than between session type. As shown in the figure, only for Learning sessions is this difference systematically greater than zero. If we prefer a NHST approach, then a sign test (for whether each session type had a median different from zero) gives Learning: $p=0.002$; Rule change $p=0.72$; Other $p=0.37$). We have added these to the legend of Figure 2e

Reviewer #3 (Remarks to the Author):

Maggi and colleagues investigated the neuronal ensemble activity patterns during inter-trial intervals (ITIs) in a rule-learning task. They quantified pairwise correlations between firing

activities of single neurons during ITIs, and compared similarities of them across different ITIs. They found that similarity of ensemble co-activity patterns during ITI increased after the animals learned the rule of the task. Furthermore, they were able to decode the information of the task events from the ensemble activity patterns. The manuscript is very clear and well written. The topic is one of great interest and it will provide large impacts for a wide audience in the fields.

We thank the reviewer for their supportive comments.

I have one major comment. In this study, the authors quantified only the neural activities during ITI periods in most of the analysis. Similarity of ensemble co-activity patterns of was assessed across different ITI periods. I think that comparisons of the ensemble activity patterns between the task trials and ITIs are also important. The analysis will provide the information whether the ensemble activity during ITI was recall of the activity in trial periods (e.g., Peyrache et al., 2009), or, it was different from the activity in task trials and represented some kind of context information (e.g. Farovik et al., 2015).

Indeed the comparison of ensemble activity between trials and ITI periods is an interesting suggestion to further investigate what was being recalled during ITIs. It could be that the recalled ensemble in the ITI reflected the ensemble activity during the trial, and thus be a memory of a potentially causal pattern of neural activity. We thus analysed the similarity between the ensemble activity in each trial and following inter-trial interval. We found that the similarity in neural correlations between the trial and its following ITI did not differ from predictions of chance-level correlations obtained from shuffled data. We found this null result across all possible combinations of session types (learning, rule change, other) and task features (outcome, direction, cue position). These results suggest the recalled ensemble in the ITI was not the trial ensemble.

We added those results in Supplementary Figure S7 and explained in the main text (lines 280 - 295).

Minor points:

1) Line #136: "To control for this, we used shuffled spike-trains to compute the expected pairwise similarities between neurons due to just the duration of each interval..."

How the authors shuffled spike-trains is not clear. Please explain it here or in Methods.

We shuffled the inter-spike intervals of each neuron independently. Lines 596-600 of the Methods now state this more fully, and hopefully more clearly.

2) Line #495: About the method to obtain a normalized spike-density function. Which periods did the authors use for calculating mean $\langle f_k \rangle$ and standard deviation σ_k ? Whole of a session, or only ITI periods?.

The Methods now clarify this: "where $\langle f_k \rangle$ is the mean f_k , and σ_k its standard deviation, taken over all the inter-trial intervals of a session." (line 565)

REVIEWERS' COMMENTS:

Reviewer #1 (Remarks to the Author):

The authors have done a wonderful job addressing the concerns raised by the reviewers. I have no further comments on this manuscript.

Reviewer #2 (Remarks to the Author):

I have read the response to reviews, and the full manuscript. All of my comments with substantively addressed, and I think this manuscript is greatly improved.

Minor comments:

Is there a spatial smoothing filter on some figures (S3b-c) but not others (S3a)?

Did they authors verify that the animals were sleeping pre/post? May have implications for learning/consolidation.

Line 92: When did the "other" sessions occur relative to the learning / rule change sessions? Before or after?

Reviewer #3 (Remarks to the Author):

The authors have addressed all of my concerns. Now I recommend to publish it in Nature Communications.

Response to reviewers

We address here the few minor comments of Reviewer 2

Reviewer #1 (Remarks to the Author):

The authors have done a wonderful job addressing the concerns raised by the reviewers. I have no further comments on this manuscript.

Reviewer #2 (Remarks to the Author):

I have read the response to reviews, and the full manuscript. All of my comments with substantively addressed, and I think this manuscript is greatly improved.

Minor comments:

Is there a spatial smoothing filter on some figures (S3b-c) but not others (S3a)?

No smoothing filter was applied to any matrix before plotting as a heatmap (eg. Fig 2). We are unclear on the specific issue spotted by the referee as FigS3b-c do not plot heatmaps.

Did they authors verify that the animals were sleeping pre/post? May have implications for learning/consolidation.

Yes, as verified in the original paper (ref. 26) the animals slept pre and post session, with bouts of slow-wave sleep in each. To make this unambiguous, we have modified the key line in the Methods to now read: "Every sleep epoch contained periods of slow-wave sleep, which were detected offline automatically from local field potential recordings (details in ref 26)".

Line 92: When did the "other" sessions occur relative to the learning / rule change sessions? Before or after?

The "other" sessions were interleaved with the learning and rule change sessions, so falling before, after, and between. In the description of the session structure in the Methods, we have added the sentence: "The learning, rule change and ``other" sessions for each rat were intermingled."

Reviewer #3 (Remarks to the Author):

The authors have addressed all of my concerns. Now I recommend to publish it in Nature Communications.